# ReLU for Inference Acceleration

## Abstract

Over the past decade, advancements in neural networks have outpaced human-level performance in a wide range of domains, including but not limited to natural language understanding and image generation. This progress has led to significantly larger networks with hundreds of billions of parameters, creating substantial computational demands. We propose the re-introduction of ReLU activation function to replace gradient-smooth alternatives during inference. We show that this can reduce computational costs while achieving minimal accuracy degradation with the help of specialized knowledge distillation training. The effectiveness of the proposed method is demonstrated by a wide variety of network architectures, covering popular applications such as image classification, object detection, and language modelling. We observed FPS improvement of 2-10% for Convolution based neural networks while observing only 1.8-2.6% accuracy degradation. The different Transformer networks demonstrated accuracy difference of approximately 1% between proposed ReLU and original GeLU networks with comparable QPS. The improvement in performance is significantly noticeable on AI accelerators like ours, with ReLU based convolution networks showcasing theoretical improvement of 41-74% compared to its SiLU based counterpart.

## 1 Introduction

In the past decade, neural networks have enjoyed unprecedented popularity and growth. Thanks to investments in data collection, hardware improvements, and architecture design, their capabilities have grown to match, and in some cases exceed that of humans in areas of natural language understanding, image generation, and more Bahrini et al. (2023)Rombach et al. (2022)Touvron et al. (2023). These growing capabilities have come with significant increases in network size and number of parameters - in some cases, exponential Thompson et al. (2023)Hoffmann et al. (2022)Bender et al. (2021). In an effort to balance these increasing computational demands, there have been significant investments in areas such as dedicated hardware (a.k.a AI accelerators), quantization, network distillation, and more.

In this paper, we propose a simple method for reducing computational cost, while observing minimal accuracy degradation, and which can be combined with quantization and hardware acceleration. Specifically, we show that though the deep learning community has largely moved away from the use of the ReLU activation function, it can be nonetheless be used as effectively, and more cheaply, as its more complicated cousins such as ELU Clevert et al. (2016), GeLU Hendrycks & Gimpel (2016), SiLU Elfwing et al. (2018), Swish Ramachandran et al. (2017), Mish Misra (2019), etc.

The ReLU activation function was a mainstay of the early- and mid-2010s deep learning network design. However, it presents some challenges during training due to its gradient discontinuity at 0, and its lack of a gradient in the negative domain. As such, many alternatives have been proposed over the past years that aim to address these by smoothing the gradient at values $\leq 0$. In particular, the GeLU and SiLU activation functions have seen tremendous popularity due to their use in transformers Devlin et al. (2019), Dosovitskiy et al. (2021) and the YOLO family of object detection networks Ge et al. (2021), Wang et al. (2023), respectively.

Activation functions are applied element-wise to the result of a linear operation, typically a matrix-matrix (GEMM) or matrix-vector (GEMV) multiplication. As such, they can be parallelized, and do not generally dominate the compute, which is mostly taken by the matrix operations. However, they still contribute some overhead. While not computationally intensive, alternative activation functions

are nonetheless more costly than the humble ReLU, requiring the calculation of exponentials, divisions, and multiplications. When viewed through the hardware acceleration lens, this can contribute a non-trivial amount of computation to a layer. For example, on our own harware, which is capable of executing a $512 \times 512$ matrix-vector multiplication (GEMV) in 512 cycles, a SiLU computation would add $10 - 15\%$ additional cycles. In contrast, due to the use of quantization and dedicated hardware, ReLU activation is effectively free.

Many applications would benefit greatly from throughput or latency improvements, while being robust to small accuracy degradation through clever system design. For example, object detection on streaming videos Li et al. (2020a) requires real time processing. Throughput increases can translate directly to reduce costs in power consumption, amount of hardware required, and maintenance. In contrast, small accuracy differences are easily mitigated by post-processing and use of object permanence across frames.

Here, we show that while alternative activation functions are critical during training to achieve high network accuracy, they are less critical during inference. With the use of knowledge distillation Hinton et al. (2015), it is possible to train an equivalent network, changing the activation, and recovering most or all of the accuracy. In contrast, training a network from scratch using the ReLU activation, will typically not perform as well.

## 2 BACKGROUND

### 2.1 ACTIVATION FUNCTIONS

Neural networks are often formed by a combinations of a parameterized linear transformation followed by a non-linear activation function to create a Universal Function Approximators Cybenko (1989), Hornik et al. (1989), Funahashi (1989), Hornik (1991). Characterising the properties of "good" activation functions is an imprecise science, there is no one-size-fits-all solution. In the early days, the activation functions were usually modelled as approximation of the biological neurons. The firing of the neurons inspired the use of Sigmoid and Tanh as network activation functions. The main problem with these activation functions is their saturation with both higher and lower inputs leading to vanishing gradients problem Pascanu et al. (2013) hurting the speed of gradient-based training of the network.

During the early- and mid-2010s, the ReLU became the most preferred activation functions in the neural network architectures due to its simplicity, sparsity and better training convergence compared to other activation functions like Sigmoid, Tanh, etc. However, the gradient discontinuity at 0 and non-existent gradients for the negative values can lead to problems like dying neuron problem, where the output of the neuron remains negative for most inputs leading to effectively zero gradient and no learning. It has been shown that the dying neuron problem significantly affects deep networks, with the proposed solution being to introduce a randomize, asymmetric initialization scheme to the neural network's weights Lu (2020). Before initialization was widely adopted, the LeakyReLU activation function was created to allow a small, non-zero gradient when the input neuron was inactive Maas (2013). LeakyReLU sacrifices the sparsity gained from ReLU for an escape from the dying ReLU problem.

Another commonly used ReLU variant is the Exponential Linear Unit (ELU) Clevert et al. (2016). The authors explain that LeakyReLU and PReLU He et al. (2015) should be preferred to ReLU for training deep neural networks to avoid dying ReLU problem and also that the presence of negative values yields a mean activation value near zero. However, one drawback to these activation functions is their undesired ability to produce large negative values. ELU combats this effect by saturating large negative values so it has the benefit of both worlds. Klambauer et al. (2017) introduce the Scaled Exponential Linear Unit (SELU) which has self-normalizing properties with a special choice of weight initialization. SELU neural networks automatically push the means and variances of activation distributions to 0 and 1 respectively. SELU-wielding neural networks outperforms many other activation functions on very deep constructions. Hendrycks & Gimpel (2016) introduce a new function called the Gaussian Error Linear Unit (GELU) which is a Gaussian cumulative distribution function. At the time of its introduction, GELU outperformed ReLU and ELU across many computer vision, natural language processing (NLP) and speech tasks. Today, GELU is the state of the art (SOTA) activation function for most transformer-based models.

Lastly, Ramachandran et al. (2017) automatically searched for activation functions using their Recurrent Neural Network Controller. Their search yielded the Swish function defined as $f(x) = x\sigma(\beta x)$ where $\beta$ can be chosen arbitrarily. Note that when $\beta = 1$, the function is called the sigmoid linear unit (SiLU). Swish consistently shows comparable or improved performance compared to ReLU, LeakyRelu, PReLU, ELU, SELU, and GELU. The advancements of activation functions has brought higher model accuracy at the cost of extra computation.

## 2.2 KNOWLEDGE DISTILLATION (KD)

In the context of machine learning, knowledge distillation (KD) is the process of using a teacher model to transfer knowledge to a student model. The most common example of KD describes the teacher model to be a wider or deeper model with the student model being narrower or shallower. KD attempts to transfer the teacher's knowledge to the student model Hinton et al. (2015), by using the prediction of the teacher model as a "soft" training label for the student. The method was evaluated using image classification task on the MNIST dataset, Deng (2012) and demonstrated promising results. Romero et al. (2015) propose a two-stage training strategy, where a hint/guided layer pair is chosen from the middle of the teacher and student models respectively. The first training stage performs KD on the hint/guided pair such that intermediate representations from the teacher are distilled to the student. The second stage of KD is performed like Hinton et al. (2015).

### 2.2.1 OBJECT DETECTION NETWORKS

These two strategies of mimicking features and soft labels guided KD for object detection models. Chen et al. (2017) incorporated a class-weighted cross entropy loss to address class imbalance issue, and adopted the teacher regression output as an upper threshold for the student to attain, rather than adhering to a rigid target. It also employed feature learning from intermediate layers to improve the training process. Guo et al. (2021) empirically validated that imitating feature maps from the Feature Pyramid Network (FPN) yields better results than feature maps from the model backbone. Li et al. (2021) demonstrated large feature map differences between student and teacher network with similar output predictions suggesting imitating complete feature map can lead to unexpected or wasted gradients. The problem of weaker anchor-object relationship especially for hard examples is addressed by Li et al. (2021) by proposing a novel, rank mimicking process to match multiple positive anchors to a certain instance and prediction guided feature imitation to focus on regions with large prediction differences. Another prominent issue with many detector is formulating the object detection multi-class classification as multiple binary classifications [Lin et al. (2017), Li et al. (2020b), Tian et al. (2019), Zhang et al. (2020)], disregarding the structural relationship between different categories. While most methods focus either on logit mimicking or feature imitation, Zheng et al. (2022) presented a novel localization distillation method inspired by viewing the bounding box regression as a probability distribution.

### 2.2.2 TRANSFORMER NETWORKS

Transformer networks are typically computationally intensive, prompting the adoption of KD as a technique for model compression. In order to reduce the size of the language model, Sanh et al. (2020) leveraged KD during pre-training phase by adopting three different losses consisting of language modeling-, KD- and cosine-distance loss. Instead of employing KD during pre-training, Sun et al. (2019) introduced Patient Knowledge Distillation (PKD) approach for task-specific training utilizing the output from the last layer as well as multiple intermediate layers of the teacher model for incremental knowledge extraction. Extending these approaches, Jiao et al. (2020) proposed a new Transformer distillation method comprising of three distillations: Transformer-layer distillation (feature mimicking for a layer's attention and hidden state outputs), Embedding-layer distillation (hidden state distillation for the embeddings), and Prediction-layer distillation (teacher prediction logits imitation). All three distillations are applied with a two-stage learning framework, during pre-training followed by task-specific fine-tuning. As the KD for transformers evolved, Lu et al. (2022) proposed a best-practice guideline based on empirical analysis of the different components like size of the student model, hyperparameters of the KD loss etc.

## 3 FORMULATION

### 3.1 RECTIFIED LINEAR ACTIVATION (RELU)

Activation functions play a pivotal role in introducing essential non-linearity to neural networks, enabling them to effectively learn an extensive array of functions. The pursuit of refining a network's capacity to model complex data patterns has led to the exploration of a diverse variety of activation functions. Among these, Rectified Linear Unit (ReLU) has been a popular choice, defined by equation 1.

$$f(x) = \begin{cases} 0, & x < 0 \\ x, & \text{if } x \geq 0 \end{cases} \tag{1}$$

Its mathematical simplicity, coupled with its semblance to biological neurons, has made it a popular selection over the years. This simplicity has also an added advantage during quantized inference, particularly for energy-efficient AI hardware.

However, ReLU is a non-smooth function and its derivative is undefined around the zero. This characteristic can lead to the 'dying ReLU' problem, wherein certain neurons effectively 'die' as they consistently output zeros. The absence of gradients for inputs less than zero prevents weight updates during training, causing these neurons to remain unchanged. This limitation has encouraged the investigation of alternative activation functions such as LeakyReLU, Exponential Linear Unit (ELU), Sigmoid Linear Unit (SiLU), Gaussian Error Linear Unit (GeLU), and others. These non-linear functions maintain smoothness and have the potential to enhance the modeling capabilities of the network. While these activation functions offer substantial advantages during network training, they can noticeably decelerate the inference process.

Neural network inference is frequently performed on energy-efficient hardware, employing data types of reduced precision. Among these, Int8 (8-bit integer) is one of the most popular data types. The precision of neural network parameters and activations are reduced to 8-bit integers, which helps save memory and computation resources while still maintaining reasonable accuracy.

The formula for quantized ReLU under INT8 quantization can be expressed as follows:

$$x^q_{ReLU} = \begin{cases} 0, & \text{if } x < 0 \\ \lfloor 255 \cdot x / \max\{X_c\} \rfloor, & x \geq 0 \end{cases} \tag{2}$$

where x is the float input to the quantized ReLU and $X_c$ is the calibration dataset. The scale factor $255/\max\{X_c\}$ is usually combined with previous or next linear layer and the resulting function for ReLU maps to $\max\{0, x\}$. The quantization process clips values to a specified maximum or minimum value, leading to no extra operations required for ReLU activation function. On the other hand, the other activation functions may require additional operations such as exponentiation, multiplication, etc. Some low bit quantization schemes can use more efficient approaches, such as precomputed lookup tables, but all these approaches result in larger latency and higher memory consumption.

### 3.2 METHOD

We propose a strategy in which intricate activation functions within a network can be substituted with ReLU. By incorporating knowledge distillation during the training of the ReLU-based network, the accuracy of the original model can be retained. The proposed algorithm is described as follows:

1. Begin by training a network employing arbitrary activation functions for the designated task. Alternatively, select a pre-trained network that already uses complex activation functions. This network is effectively considered as a teacher model.

2. Transition by replacing the intricate activation functions with ReLU, effectively designating this transformed network as the student model.

3. Initialize both the teacher and student models using the trained weights of the teacher model.

4. Employ task-specific loss functions and introduce distillation loss during the training of the student model. This combination of loss functions enables the student model to not only learn from the original task but also leverage insights distilled from the teacher model.

By following this sequence of steps, we can effectively replace complex activation functions with ReLU while maintaining the accuracy of the network. A diverse array of distillation loss functions is available, tailored to various models and tasks. Next, we describe three different distillation losses, each aligned with a different task.

### 3.2.1 IMAGE CLASSIFICATION

For image classification networks built upon transformer architectures, we employed the distillation loss introduced by Touvron et al. (2021) as formulated by the following equation 3.

$$\mathcal{L}_{global}^{hardDistill} = \frac{1}{2}\mathcal{L}_{CE}(\psi(Z^s), y) + \frac{1}{2}\mathcal{L}_{CE}(\psi(Z^s), y^t) \tag{3}$$

Where:

- $Z^s$ is the logits of the student model, $y$ is ground truth label and $y^t = \arg\max_c Z^t(c)$ be the hard decision of teacher model with $Z^t$ representing logits of teacher model.
- $\psi$ is softmax function and $\mathcal{L}_{CE}$ is cross-entropy loss.

### 3.2.2 OBJECT DETECTION

CNN based object detection networks are trained using channel-wise knowledge distillation loss described by equation 4 adopted from Shu et al. (2021).

$$\mathcal{L}^{Distill} = \frac{1}{C}\sum_{c=1}^{C}\sum_{i=1}^{W \cdot H}\psi(f_{c,i}^t)\log\left[\frac{\psi(f_{c,i}^t)}{\psi(f_{c,i}^s)}\right] \tag{4}$$

Where:

- $f^s$ and $f^t$ are the feature activation maps of the student and teacher models respectively.
- $c = 1, 2, ..., C$ indexes the channel and $i$ indexes the spatial location of the channel.
- $\psi$ is softmax function.

### 3.2.3 LANGUAGE MODEL

KL-Divergence loss is employed for Distillation of language models and it is defined by equation 5

$$\mathcal{L}_{KLD}^{Distill} = \psi(Z^t)\log\left[\frac{\psi(Z^t)}{\psi(Z^s)}\right] \tag{5}$$

where $Z^s$ and $Z^t$ are the logits of the student and teacher model respectively.

### 3.2.4 SMOOTH ACTIVATION TRANSITION (SAT)

While ReLU demonstrates benefits during the inference process, it often falls behind during training of modeling intricate data patterns. We recognise that many of the complex activation functions like SiLU or GeLU are largely similar to ReLU while differing around the zero-point as shown by figure 1. This similarity can help in transitioning smoothly from complex activation function to simpler efficient activation function. We proposed to modify the original activation function to be a combination of original function as well ReLU activation function with a weighted sum defined by equation 6.

$$f_{SAT}(x) = \gamma f(x) + (1 - \gamma)\max(0, x) \tag{6}$$

$f(x)$ is the original activation function and $\gamma$ is a hyper-parameter modified from 1 to 0 during the course of the training.

This method of smooth transitioning from original function to ReLU can help the network to stay around the local optimal point while moving to a new and efficient non-linear activation function.

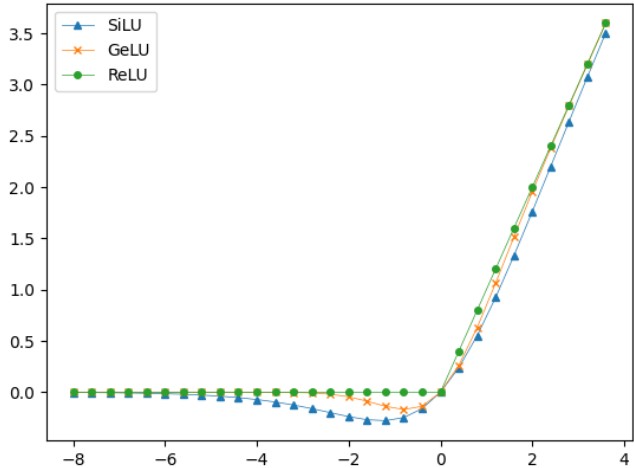

Figure 1: Activation functions: ReLU, SiLU and GeLU

## 4 RESULTS

This section demonstrate accuracy and inference speed impact of ReLU based networks for different networks performing distinct tasks.

### 4.1 IMPLEMENTATION DETAILS

We made modifications to the original code by replacing the activation function with ReLU and introduced knowledge distillation where it was not initially implemented. We trained ReLU-based networks using two distinct sets of initial weights: one with the original baseline model weights, and the other with randomly initialized weights. Throughout this process, we maintained the original network training parameters and augmentation techniques, with adjustments made solely to the learning rate, batch size, and the number of epochs as necessary. Subsequently, we carried out an inference acceleration analysis on RTX-4090 GPUs for the trained networks by converting them into INT8 TensorRT engines.

### 4.2 PERFORMANCE COMPARISON

We present a comparative analysis of speed and accuracy between ReLU-based networks and the original SiLU model for single-stage object detectors DamoYolo[1], YoloXL[2], and YoloV7[3], as shown in Table 1. The networks are trained using knowledge distillation loss described in section 3.2.2 Accuracy was measured on the MSCOCO validation set using the FP32 data type, while FPS was measured using TensorRT INT8 engines. In the table, 'B' represents baseline models with pretrained weights, whereas 'KD' and 'B+KD' represent trained networks using knowledge distillation, starting from randomly initialized and baseline model weights, respectively. Our observations indicate that while the accuracy of the network decreased by 1.8-2.6%, the FPS improved by 2-10%.

In order to further improve accuracy of the relu based model, we trained the models using smooth activation transition method described in section . The models are trained by starting from the baseline SiLU model weights and modifying the $\gamma$ value from 1 to 0 with the interval of 0.1. The results are presented in table with almost equivalent or slightly poor performance compared to simply train-

---

[1]https://github.com/tinyvision/DAMO-YOLO

[2]https://github.com/Megvii-BaseDetection/YOLOX

[3]https://github.com/WongKinYiu/yolov7

| Method | Activation | FPS | $AP$ | $AP^{50}$ | $AP^{75}$ | $AP^S$ | $AP^M$ | $AP^L$ |
|---|---|---|---|---|---|---|---|---|
| DamoYolo(M)-B | SiLU | 968 | 50.0 | 66.9 | 54.7 | 30.5 | 54.8 | 67.6 |
| DamoYolo(M)-KD | ReLU | 999 | 44.3 | 60.8 | 48.4 | 25.2 | 48.5 | 62.4 |
| DamoYolo(M)-B+KD | ReLU | 999 | 48.7 | 65.8 | 53.2 | 29.7 | 53.5 | 65.9 |
| DamoYolo(M)-B+KD-SAT | ReLU | 999 | 48.5 | 65.6 | 52.8 | 29.2 | 53.3 | 65.9 |
| YoloX(L)-B | SiLU | 802 | 49.7 | 68.0 | 54.0 | 32.3 | 54.9 | 65.1 |
| YoloX(L)-KD | ReLU | 844 | 45.9 | 64.4 | 49.8 | 28.1 | 50.4 | 60.0 |
| YoloX(L)-B+KD | ReLU | 844 | 48.9 | 67.5 | 53.1 | 31.6 | 53.5 | 64.2 |
| YoloX(L)-B+KD-SAT | ReLU | 844 | 48.5 | 67.1 | 52.5 | 30.9 | 52.1 | 63.6 |
| YoloV7-B | SiLU | 742 | 51.2 | 69.7 | 55.9 | 31.8 | 55.5 | 65.0 |
| YoloV7-KD | ReLU | 810 | 46.4 | 64.2 | 50.6 | 30.0 | 51.1 | 60.2 |
| YoloV7-B+KD | ReLU | 810 | 50.3 | 68.6 | 54.6 | 33.3 | 54.9 | 66.0 |

Table 1: Performance comparison of ReLU vs SiLU networks for state-of-the-art object detectors on COCO validation dataset. B indicates baseline weights, KD is used for training from random weights with knowledge distillation, B+KD indicates training from pre-trained weights with knowledge distillation, and B+KD-SAT denotes object detectors trained using smooth activation transition method. FPS is measured using TensorRT engine in INT8 on RTX 4090 GPUs.

| Method | Activation | QPS | % Exact Match | % F1 |
|---|---|---|---|---|
| Bert(Base-SQuAD)-B | GeLU | 2003 | 81.48 | 88.69 |
| Bert(Base-SQuAD)-KD | ReLU | 1977 | 82.47 | 89.39 |
| Bert(Base-SQuAD)-B+KD | ReLU | 1977 | 82.51 | 89.41 |

Table 2: Comparison of ReLU vs SiLU network performance for the BERT-Base Model SQuAD task. FPS is measured using TensorRT engine in INT8 on RTX 4090 GPUs

ing ReLU model with baseline weight. The different variations of hyperparameters may result into better accuracy which could not be explored due to limited time and resources.

We also conducted an evaluation on the impact of replacing GeLU activation with ReLU activation in transformer-based networks. Specifically, we trained a Bert-Base[4] network with ReLU activation for the SQuAD task, employing knowledge distillation loss as described in the section 3.2.3. The results, as presented in Table 2, demonstrate that the ReLU-based network outperforms the original GeLU-based network in terms of both exact match and F1 score.

Furthermore, we extended our analysis by training another transformer-based network, DeiT[5], with ReLU activation for an image classification task, utilizing knowledge distillation loss as described in the section 3.2.1. The results, showcased in Table 3, reveal comparable performance between the baseline and ReLU-based networks.

### 4.2.1 AI ACCELERATOR SPEED COMPARISON

In order to analyse the impact of the simplified computation of ReLU activation on dedicated AI hardware, we theorize the performance on one of hardware for popular object detection networks DamoYolo and YoloXL. The results presented in table 4 demonstrates significant FPS improvement of 41-74% and latency reduction of 37%. We observed the cycle count difference between ReLU and SiLU can be of the order of 100x validating higher inference of ReLU networks.

---

[4] https://github.com/NVIDIA/DeepLearningExamples/tree/master/PyTorch/LanguageModeling/BERT
[5] https://github.com/facebookresearch/deit

| Method | Activation | Acc1 | Acc@5 |
|---|---|---|---|
| DeiT(Small)-B | GeLU | 81.2 | 95.4 |
| DeiT(Small)-KD | ReLU | 80.5 | 95.0 |
| DeiT(Small)-B+KD | ReLU | 81.0 | 95.3 |
| DeiT(Base)-B | GeLU | 83.4 | 96.5 |
| DeiT(Base)-KD | ReLU | - | - |
| DeiT(Base)-B+KD | ReLU | 82.9 | 96.2 |

Table 3: Comparison of ReLU vs SiLU network performance for the transformer based image classifier DeiT models on ImageNet dataset.

| Method | Activation | Latency | FPS |
|---|---|---|---|
| DamoYolo(M)-B | SiLU | 9.05 | 320 |
| DamoYolo(M)-B+KD | ReLU | 5.69 | 559 |
| YoloXL-B | SiLU | 15.04 | 221 |
| YoloXL-B+KD | ReLU | 9.42 | 313 |

Table 4: Comparison of ReLU vs SiLU network inference speed on dedicated AI Accelerator

## 5 SUMMARY

In this study, we demonstrated the effectiveness of ReLU based networks on variety of applications and network architecture with minimum accuracy degradation. For convolution networks, when employing pre-trained baseline weights, the observed accuracy degradation ranged from 1.8% to 2.6%, significantly outperforming the 7.6% to 11.4% degradation experienced when initializing with random weights. The minor accuracy degradation of ReLU network with baseline weights brought 2-10% FPS improvements on GPU and 41-74% FPS improvements on our dedicated AI hardware. In case of transformer network, Bert-Base, ReLU network has shown accuracy improvement of 1.2% when starting from baseline pre-trained as well as random initial weights. However, considering the modern GPU performance is highly optimized for transformers, transitioning to the ReLU activation function can reduce the level of optimization, resulting in a 1.3% reduction in QPS when compared to the GeLU network on GPU. Additionally, DeiT networks exhibited only a slight accuracy drop of less than 1% when using both baseline and randomly initialized weights.With comparable accuracy achieved by ReLU and GELU-based transformer networks, along with only minor accuracy degradation observed in Convolution-based networks, it becomes evident that ReLU-based networks offer a viable option for the inference accelerators, offering significant speedup benefits.

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
