# OpenReview forum: "ReLU for Inference Acceleration"
_ICLR.cc/2024/Conference — ICLR 2024 Conference Withdrawn Submission_

### Official Review · Reviewer_RjcT · 2023-10-24

**Soundness:** 2 fair
**Presentation:** 2 fair
**Contribution:** 2 fair
**Rating:** 5
**Confidence:** 4

**Summary:**

This paper demonstrates to use the ReLU as an activation function for inference. While more complex activation functions are beneficial during training, they incur overhead during inference. With knowledge distillation, it is possible to reduce the accuracy loss when using the ReLU. The results implemented on different benchmarks show tradeoffs between accuracy and efficiency.

**Strengths:**

1. The proposed method consists of a simple but effective idea.

2. The experiments have been conducted on a wide set of benchmarks.

**Weaknesses:**

There are some suggestions for improvements. Please refer to the questions below.

**Questions:**

1. Please clarify what is the state-of-the-art of prior work that use ReLU for inference on networks that were trained with more complex activation functions. What are the limitations and challenges of the related work? What are the innovations made in these paper?

2. It is recommended to compare the proposed solution using ReLU with approximate implementation of the complex original activation functions to explore the tradeoffs between accuracy and efficiency.

3. Besides FPS and latency, it would be interesting to compare the results in terms of energy consumption between different activation functions.

---

### Official Review · Reviewer_jbaK · 2023-10-31

**Soundness:** 3 good
**Presentation:** 2 fair
**Contribution:** 2 fair
**Rating:** 3
**Confidence:** 5

**Summary:**

This paper proposes to use ReLU, instead of other complicated activation functions, in different transformer models for higher efficiency during inference. During training, this method uses knowledge distillation to mitigate the accuracy drop and can reduce computational costs with minimal accuracy degradation.

**Strengths:**

- This paper demonstrates the effectiveness of the proposed on different models, covering different tasks such as classification, object detection, and language modeling.

**Weaknesses:**

- The technical novelty of this paper is limited. The ReLU is more efficient than GeLU/SiLU is well known. And in ConvNext[1], they have discussed the model accuracy with ReLU and GeLU. Only replacing the activation function to ReLU, and without more theoretical analysis, does not contribute much.
- The knowledge methods used in this paper are all existing methods.
- This method need to train more with knowledge methods, which is not efficient for model training.

[1] Zhuang Liu, et al. A ConvNet for the 2020s. CVPR, 2022

**Questions:**

Typo
- In abstract, “language modelling” -> “language modeling”

---

### Official Review · Reviewer_9ug7 · 2023-10-31

**Soundness:** 2 fair
**Presentation:** 3 good
**Contribution:** 2 fair
**Rating:** 3
**Confidence:** 4

**Summary:**

The authors propose a series of methods in order to replace modern activation functions with ReLU after training while minimizing quality loss. Their method is based on a combination of knowledge distillation and smooth interpolation from the original function to ReLU.

**Strengths:**

The paper was well written and clear aside from a few limited places I note below.

**Weaknesses:**

I think there are two primary weaknesses in this paper. First, I did not find the motivation for using ReLU to be very compelling. The author’s case seems to be primarily predicated on an “AI accelerator” which remains unnamed. The paper provides essentially no detail on this alternative hardware accelerator other than some cycle count costs of matrix multiplication and SiLU in the introduction. The authors refer to this accelerator as “our” hardware in multiple places, which suggests to me that they may have omitted details to maintain anonymity. I think the paper would be much stronger if the authors re-wrote to include details on the accelerator of interest and avoided claiming it as “ours” to maintain anonymity (although it is too late for this change in this review cycle, obviously).

Second, the results on Nvidia GPUs mixed and confusing in places. The quality loss/runtime improvements in Table 1 seem to suggest this is not a very good tradeoff. For example, 1.5% lower AP with DamoYolo for a 3% improvement in runtime? 1.5% accuracy loss is somewhat significant and I would expect that training a slightly smaller model would yield larger runtime improvements for the same quality. I also found the results in Table 2 somewhat contradictory to the paper up to that point. Switching from GeLU to ReLU appears to make the model slower? But it also increases the model quality?

**Questions:**

In section 4.2.1, what does “we theorize the performance on one of hardware” mean?

---

### Official Review · Reviewer_SAkP · 2023-11-01

**Soundness:** 3 good
**Presentation:** 3 good
**Contribution:** 2 fair
**Rating:** 5
**Confidence:** 4

**Summary:**

This paper proposes using the ReLU activation function in place of more complex activations like SiLU during inference to improve speed and efficiency. The key idea is that while smooth activations like SiLU are important for training, ReLU can be used during inference with minimal accuracy loss if knowledge distillation is used during training. The method is evaluated on image classification, object detection, and language models.

**Strengths:**

* The proposed method is simple but effective - replacing activation functions with ReLU improves inference speed across models and tasks with small accuracy drops.
* The paper thoroughly evaluates the approach on major model types - CNNs, detectors like YOLO, and transformers. The consistent gains show the broad applicability.
* Detailed experiments quantify the tradeoff between accuracy and speed. Up to 10% higher FPS is achieved on GPUs and even larger gains on specialized hardware.
* The method can be combined with other optimizations like quantization for greater efficiency.

**Weaknesses:**

* The rationale behind initializing the student model (which has a ReLU activation function) with the teacher's weights is unclear. Does it aid in optimization, or does it provide the student model with a higher initial accuracy?
* While all the components used in this study are well-established and the proposed method appears to be a straightforward amalgamation of techniques, it would be interesting to know the generality of the method. In which scenarios might the method falter and in which might it produce excellent performance?
* The results pertaining to specialized hardware, though promising, are merely theoretical estimations. Practical implementation could potentially uncover additional trade-offs.

**Questions:**

see weakness

---

### Author Response · Authors · 2023-11-17
**Thank you very much for your time and constructive feedback**

Thank you for the detailed review and constructive comments. While we don't align on every point, we sincerely appreciate your time and feedback.